# Long-Period Fiber Grating Sensor Based on a Conductive Polymer Functional Layer

**DOI:** 10.3390/polym12092023

**Published:** 2020-09-04

**Authors:** Ching-Yu Hsu, Chia-Chin Chiang, Hsin-Yi Wen, Jian-Jie Weng, Jing-Lun Chen, Tao-Hsing Chen, Ya-Hui Chen

**Affiliations:** 1Department of Marine Mechanical Engineering, R.O.C. Naval Academy, Kaohsiung 81345, Taiwan; cyhsu@mail.cna.edu.tw; 2Department of Mechanical Engineering, National Kaohsiung University of Science and Technology, Kaohsiung 80778, Taiwan; ccchiang@nkust.edu.tw (C.-C.C.); hywen@nkust.edu.tw (H.-Y.W.); F107142128@nkust.edu.tw (J.-J.W.); chinesetaipeibasketball@gmail.com (J.-L.C.); thchen@nkust.edu.tw (T.-H.C.)

**Keywords:** long-period fiber grating (LPFG), laser-assisted-etching LPFG (LLPFG), fiber sensor, PEDOT:PSS, temperature detection

## Abstract

A temperature sensor was fabricated with a functional conductive poly(3,4-ethylenedioxythiophene) polystyrene sulfonate (PEDOT:PSS) coating on a long-period fiber grating (LPFG). The LPFG was fabricated by laser-assisted wet-chemical etching for controlling the grating depth of the LPFG after the treated surface of an optical fiber was inscribed by laser light. The functional conductive polymer acts as a temperature sustained sensing layer and enhances the grating depth of the LPFG sensor as a strain buffer at various temperatures. The sensor was subjected to three cycles of temperature measurement to investigate the sensor’s wavelength shift and energy loss when exposed to temperatures between 30 and 100 °C. Results showed that the sensor’s average wavelength sensitivity and its linearity were 0.052 nm/°C and 99%, respectively; average transmission sensitivity and linearity were 0.048 (dB/°C) and 95%, respectively.

## 1. Introduction

Recently, there has been considerable interest in fiber grating refractive index (RI) sensors because of their high sensitivity, efficiency, and light weight. In particular, close attention has been paid to long-period fiber gratings (LPFGs), particularly to the way in which light at the resonant wavelength of an LPFG is coupled from the guided to the cladding modes, which causes energy loss depending on the difference between the RI of the guided and cladding modes. For this reason, the previous study proposes a laser-assisted-etching LPFG (LLPFG) sensor fabricated using an excimer laser and metal photomasks, examines the effects of temperature on wavelength drift and transmission loss, and explores the use of LLPFG sensors for temperature sensing [1,2].

This type of functional coated layer is frequently employed as a feature in optical fiber sensors. A functional group of materials portrays a chemical or physical reaction with a purpose, for example, a small molecular structure grown layer-by-layer via deposition, or solution coating manufacture of a polymolecular structure by sol–gel or dip coating [3,4,5].

In 2008, Ding et al. [6] utilized reactive ion etching to etch a long-period grating structure on a tapered optical fiber; the etched structure had a length of 11 mm, a fiber diameter of 15 µm, and a long-period distance of 200 µm. In addition to the fabrication of a long-period grating structure through reactive ion etching, that study also examined the relationship between the tapered optical fiber’s resonant wavelength and etched fiber diameter. Through a comparison of the measured and calculated transmission spectrum values, it was found that multimodal coupling occurred only between two neighboring tapers and at the boundaries of the LPFGs. In the same year, Zhao et al. [7] inscribed a long-period structure on a photonic crystal fiber and used it to measure strain. The measurement method involved adhering one end of the long-period structure to a fixed platform and the other end to a mobile micromotion platform such that the micromotion platform was used to control displacement and thus measure strain at a resolution of 0.3 με. 

In 2012, Chiang et al. [8] proposed an LPFG fabricated using inductively coupled ion etching. This primarily involved the use of photolithography to imprint a periodic structure on a photoresist to be imposed on an optical fiber. The coupled plasma ion etching process was then performed, during which the photoresist served as a barrier layer. An etch rate of 0.35 μm per minute was achieved. Subsequently, the structure was placed in sulfuric acid to remove the photoresist and obtain LPFGs with certain periods that had stable verticality. A comparison with wet etching results indicated that the coupled plasma ion etching process offered superior etching stability and accuracy. 

Given the broad potential of temperature transducers across various commercial and industrial fields, Singh et al. [9] proposed an LPFG temperature transducer in 2014. This LPFG temperature transducer can respond to the various peak resonance wavelength shifts corresponding to the various attenuation bands of transmission spectra. After studying the effects of temperature on the various attenuation bands of the LPFG, a highly sensitive measuring device was developed. Subsequently, after monitoring the wavelength shift of each peak resonant wavelength in response to temperature increments of 20 °C, it was found that, with respect to the temperature sensitivity of various attenuation bands, the LPFG had a grating period of 280 μm within the wavelength range of 1.1–1.7 nm (0 to 100 °C). In 2015, Wu et al. [10] proposed a temperature measurement experiment that involved the use of an LPFG with a nanostructure. In this experiment, an increase in temperature was accompanied by the red-shifting of wavelength positions and a gradual increase in loss; with a diameter of 45 µm and period of 620 µm, the wavelength’s temperature sensitivity was 0.0693 nm/°C, and the sensitivity of loss in response to temperature was 0.043 dB/°C. 

In 2015, Bai et al. [11] proposed the fabrication of long-period structures using a welding machine and a cutting head. In that study, a welding machine (eccentricity was set to 3 μm) was used to weld three segments of LPFGs with a period of 586 μm. When temperatures from 20 to 800 °C were measured, the LPFG was found to have a loss sensitivity of 0.00123 dB/°C and a wavelength sensitivity of 0.0977 nm/°C; however, from 800 to 1000 °C, the LPFG was revealed to have a loss sensitivity of 0.0199 dB/°C and a wavelength sensitivity of 0.2652 nm/°C. In 2018, Zhang et al. [11] proposed the use of a femtosecond-calibrated CO_2_ laser to fabricate an all-fiber dual-parameter transducer with a cascaded long-period grating, and managed to produce results that contributed to the theoretical and experimental advancement of this field. The resulting LPFGs had a resonant wavelength of 1557.80 nm and 1590.88 nm; within a strain range of 0 to 400 με, the LPFGs fabricated by CO_2_ laser (C-LPFG) and by femtosecond laser (F-LPFG) achieved a strain sensitivity of −7.2 and −1.6 pm/με, respectively. Within a temperature range of 30 to 70 °C, the C-LPFG and F-LPFG achieved values of −41.1 and −21.2 pm/°C. After the characteristics of the resonant wavelengths had been analyzed, it was found that the transducer could effectively perform two-parameter measurements, giving it broad applicative potential and enhancing its value as a research reference. 

In 2019, Wang et al. [12,13] designed a high-sensitivity temperature transducer made from tapered multimode chalcogenide fiber that had LPFGs. From a theoretical perspective, they examined these LPFGs’ transmission characteristics, the temperature sensitivity of their cladding modes, and the changes to their diameters and surrounding refractive indices. The results proved that the LPFGs’ temperature sensitivity values could be effectively increased by reducing their diameters. When a particular LP_02_ fiber had a minimum diameter of 75 μm, the corresponding LPFG’s temperature sensitivity per 1.55 μm was calculated to be 1.89 nm/°C. When the grating period of the LPFG was selected at its dispersion turning point, its temperature sensitivity per 1.55 μm reached a maximum absolute value of 15.2 nm/°C, which was approximately 120 times the sensitivity achieved by a tapered silicon dioxide LPFG. In 2020, Wen et al. [14] proposed a novel square-wave double-sided long-period fiber grating structure through a KrF 248-nm excimer and an aqueous solution of acid. The square-wave period grating structure was caused by an etching rate on the optical fiber surface after a laser-assisted wet etching process. Average resonant wavelength/temperature ratio of the sensor was 0.054 nm/°C, while the transmission loss/temperature ratio was 0.038 dB/°C. This study proposes the application of a conductive polymer in a temperature detector based on an LPFG. The interesting concept of an optical fiber sensor via a conductive polymer functional layer is novel and unpublicized.

## 2. Theory

Another term for LPFG is transmission fiber grating. Any LPFG has a period of approximately 100 to 1000 μm. When light is being transmitted through an optical fiber, the periodic grating structure can be utilized to determine the phase-matching conditions for the LPFG’s resonant wavelengths using the following equation [15]:(1)$$\lambda =\Lambda \left(n_{\mathrm{core}}^{\mathrm{eff}}-n_{\mathrm{cladding}}^{\mathrm{eff}}\right)$$
where λ is the resonant wavelength position, Λ is the grating period, $n_{\mathrm{core}}^{\mathrm{eff}}$ is the core’s effective refractive index, and $n_{\mathrm{cladding}}^{\mathrm{eff}}$ is the cladding’s effective refractive index.

When light passes through the grating region and causes the periodic grating’s refractive index to change, the LPFG’s transmission loss is defined as follows:(2)$$T=\;\frac{1}{1+\frac{K^{\mathrm{de}}}{K_{\mathrm{co}-\mathrm{cl}}^{\mathrm{ac}}}}\cos^{2}\left(\sqrt{{\left(K_{\mathrm{co}-\mathrm{cl}}^{\mathrm{ac}}\right)}^{2}+{\left(K^{\mathrm{dc}}\right)}^{2}}L\right)$$
where $k^{\mathrm{dc}}$, $k_{\mathrm{co}-\mathrm{co}}^{\mathrm{dc}}$, and $k_{\mathrm{cl}-\mathrm{cl}}^{\mathrm{dc}}$ are the LPFG’s coupling coefficients, and $k^{\mathrm{dc}}=$
$k_{\mathrm{co}-\mathrm{co}}^{\mathrm{dc}}=$
$k_{\mathrm{cl}-\mathrm{cl}}^{\mathrm{dc}}=0$. Thus, Equation (2) can be simplified as follows:(3)$$T=\cos^{2}\left(k_{\mathrm{co}-\mathrm{cl}}^{\mathrm{ac}}L\right)$$

Because the transmission loss of light being transmitted through an LPFG is a cosine-squared function, the magnitude of the LPFG’s transmission loss is determined by the coupling coefficient $k_{\mathrm{co}-\mathrm{cl}}^{\mathrm{ac}}$ and grating length (L).

The sensor’s grating period changes in response to thermal expansion or contraction, which, in turn, causes the coupling coefficients to change. The sensing response of the LPFG’s grating period was investigated in this experiment on the basis of this principle.

## 3. Material and Methods

### 3.1. Fabrication and Laser-Assisted Etching of LPFG

In this experiment, it was necessary to first fabricate the sensor component in the optical fiber (see Figure 1). First, 3 cm of protective sheathing was stripped from the middle section of a single mode optical fiber (125 μm in diameter), after which the optical fiber was adhered to a metal plate with a 620-μm period. The plate was then secured to a triaxial micromotion stage as shown in Figure 1a. The stage was moved to allow the laser to scan the periodic metal plate onto the optical fiber. The excimer laser was focused on the surface of the optical fiber and set to 12 mJ as the micromotion stage was moved at a speed of 0.2 mm/s, which allowed for the grating to be created. Next, the period-modified optical fiber underwent wet etching, during which buffered oxide etch (BOE) was used, and the optical fiber was etched to a diameter of 65 μm (see Figure 1b). Because defects on the surface affect the etch rate, the grating grooves in the outer section were deepened as illustrated in Figure 2a.

### 3.2. Tensile Testing

For this test, one end of the LLPFG sensor, which had been fabricated through laser-assisted etching, was connected to an optical spectrum analyzer (OSA) (Anritsu, MS9710C, Kanagawa Prefecture, Japan); the other end was connected to a superluminescent diode. These two ends were then secured to the micromotion stage and load cell; the horizontal positions of the micromotion stage and load cell were then adjusted. Tensile testing was conducted by manually rotating the micromotion stage such that an axial force was created to stretch the sensor. The tensile force was increased at increments of 0.0098 N, and the changes to the OSA’s wavelength readings were observed, with a waveform rebound indicating that the sensor’s optimal sensitivity had been reached. The experimental scheme is illustrated in Figure 3.

### 3.3. PEDOT:PSS Coating Process and Sensor Encapsulation

During the encapsulation process, the LLPFG sensor was placed above a glass dish, and a corona treater was used was to induce a gas breakdown; specifically, a dissociation reaction was achieved by using the corona treater to perform a tip discharge at the upper section of the sensor structure such that a concentrated electric field was formed at the tip. After the aforementioned process had been completed, the sensor was placed in a cubic quartz cell where it was then coated with PEDOT:PSS.

One round of coating took approximately 10 min, during which the sensor was heated by a furnace to allow the PEDOT:PSS to spread evenly across the sensor. The coating of the sensor was completed after this step had been repeated five times. After the encapsulation process had been completed, tensile testing was performed to allow the sensor to achieve maximum sensitivity, and the force applied to the micromotion stage was reduced by 0.0098–0.00196 N to prepare it for the encapsulation process. A quartz slide was secured to the middle of an elevating platform, and the sensor was placed in the central space. One end of the sensor was secured by adhering it to one end of the quartz slide using ultraviolet (UV) glue (Technology Co., Ltd. Taiwan) and exposing it to UV light for 20 s. The micromotion stage was then used to apply tension to the other end of the sensor until optimal sensitivity (as indicated by the tensile test) was achieved. At this point, the sensor’s other end was secured by using UV glue and exposing the sensor to UV light for 20 s. The schematic for this process is depicted in Figure 4.

A scanning electron microscope (SEM) (Philips XL 30, Royal Dutch Philips Electronics Ltd., Amsterdam, Nederland) image and a microscan photo of the optical fiber with an incident laser beam at the grating depth during wet etching appear as Figure 2a,b, respectively. The LPFG, with a period of 620 μm, was scanned using a KrF excimer laser; a 21 μm depth in the grating areas was produced through the reaction with the acid solution during wet etching. Figure 2c displays a microscan photo of the optical fiber with a polymer coating; the image indicates that the surface roughness of the optical fiber was uniform.

Figure 5 presents a schematic of the molecular structure of PSS. PEDOT:PSS is a water-dispersible form of conducting PEDOT dopant with water-soluble PSS. Stretchable conducting layers (including electronic skin and textiles, actuators, artificial muscles, sensors, and the like) are favored for next-generation consumer electronics owing to their excellent processability and appropriate conductivity, flexibility, and stretchability.

### 3.4. Temperature Testing

The PEDOT:PSS-coated LLPFG sensor was first heated in a furnace. One end of the sensor was connected to an OSA; the other end was connected to a superluminescent diode. The two ends were then secured to the micromotion stage and load cell, after which the horizontal positions of the micromotion stage and load cell were adjusted primarily to prevent the sensor from bending. Next, a thermocouple was used to monitor the temperature change inside the furnace in real time, and data acquisition tools (LabVIEW 2011) were used to collect the temperature data that were generated. The schematic for this process is displayed as Figure 3. When an LPFG is under loadings or external environmental effects, such as forces, strains, torsions, or temperatures, the transmission spectra change with wavelength shift and resonance-attenuation loss. Hence, this implies that the LPFG is also sensitive to external environmental perturbations such as temperature and vibration. To reduce the effects of temperature and vibration on the sensor, the experimental plant was controlled with damped optical tables to avoid the effects of any vibration change, but artificial operation vibration is inevitable.

## 4. Results and Discussion

The PEDOT:PSS-coated LLPFG sensor was subjected to temperatures ranging from 30 to 100 °C to analyze its sensitivity. The sensor was created with the use of an excimer laser and an iron grating with a 620-μm period, after which BOE was used to wet etch the sensor to a diameter of 65 μm. Following this, the LLPFG sensor underwent tensile testing, during which it was subjected to tensile forces ranging from 0 to 0.0882 N; it achieved its optimal sensitivity at 0.0882 N. Finally, PEDOT:PSS coating and encapsulation processes were performed.

The three-cycle temperature test spectrogram is shown in Figure 6. During temperature testing, the sensor was exposed to a gradual increase in temperature from 30 to 100 °C over a period of 1 h, and furnace cooling was then used to return the temperature to its original level under stable conditions. As graphed in Figure 6, the increase in temperature was accompanied by a gradual shift of the wavelength position to a longer wavelength, as well as a gradual decline of transmission loss. Table 1 also indicates that the wavelength shift was approximately 4 nm. The changes in wavelength shift and transmission loss are listed in Table 1.

Wavelength losses observed over three temperature test cycles are graphed in Figure 7. This figure indicates that the sensor’s average wavelength sensitivity and average transmission sensitivity were 0.052 nm/°C and 0.048 dB/°C, respectively (also see Table 2). As listed in Table 3, the LLPFG exhibited accurate linearity for wavelength shift and transmission loss, with its average wavelength shift linearity and average transmission loss linearity being 0.99 and 0.95, respectively. These results also indicate that Equation (1) was fulfilled because the optical fiber’s refractive indices ($n_{\mathrm{core}}^{\mathrm{eff}}$ and $n_{\mathrm{cladding}}^{\mathrm{eff}}$) and wavelength position increased in response to temperature changes. The transmittance of an LLPFG can be expressed with the coupling coefficient between the core and cladding and the grating length. The length of the LLPFG is determined by the overlap integral of the core and cladding modes and by the amplitude of the periodic modulation of the mode propagation constants. When the thermal expansion is applied, the value of $k_{\mathrm{co}-\mathrm{cl}}^{\mathrm{ac}}$ in Equation (3) will change according to the elastic-optic effect. Therefore, the transmittance can be tuned by changing the external expansion. Therefore, Equation (3) was also fulfilled because the grating period changed in response to thermal expansion and contraction, which, in turn, led to variations in the coupling coefficients and gradual increases or decreases in transmission loss.

We also conducted control experiments (the temperature sensitivity of an LPFG without PEDOT:PSS). Wavelength shift and transmission loss sensitivity over three temperature test cycles under LPFG (D = 65 μm, Λ = 620 μm) coated with PEDOT:PSS and LPFG (D = 65 μm, Λ = 620 μm) without PEDOT:PSS (Figure 8) are similar. However, the LPFG sensor without PEDOT:PSS frequently broke during temperature detection. The polymer plays the role of a protective layer for the sensor, and the polymer can expand and contract with temperature changes during temperature measurement. Therefore, the result can reach similar sensitivity.

## 5. Conclusions

Our study proposed a PEDOT:PSS-coated LLPFG sensor with a diameter of 65 μm and a period of 620 μm and subjected it to three temperature test cycles with the aim of investigating the sensor’s wavelength shifts and transmission losses when exposed to temperatures between 30 and 100 °C. The experimental results proved that when the PEDOT:PSS-coated LLPFG sensor was subjected to a rising temperature, its wavelength position gradually shifted to a longer wavelength (the average wavelength shift was 4 nm) and its transmission loss was reduced. As demonstrated in the chart and tables above, the sensor’s average wavelength sensitivity and average wavelength shift linearity were 0.052 nm/°C and 99%, respectively; its average transmission sensitivity and average transmission loss linearity were 0.048 (dB/°C) and 95%, respectively. These experimental results indicate that this type of PEDOT:PSS-coated LLPFG sensor offers high sensitivity and reproducibility, and for this reason, a coating consisting of PEDOT:PSS and other polymeric materials could be applied to LLPFG sensors to increase their sensitivity and to enable various parameters to be designed and measured.

## Figures and Tables

**Figure 1 polymers-12-02023-f001:**
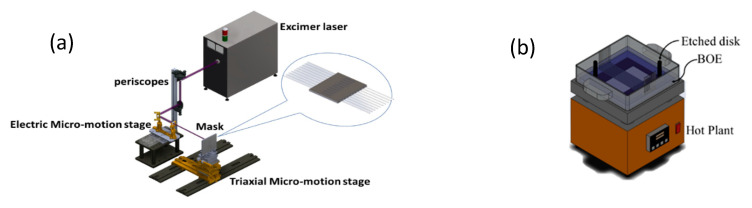
(**a**) KrF excimer laser setup used to assist chemical etching form period structure (**b**) process of chemical etching.

**Figure 2 polymers-12-02023-f002:**
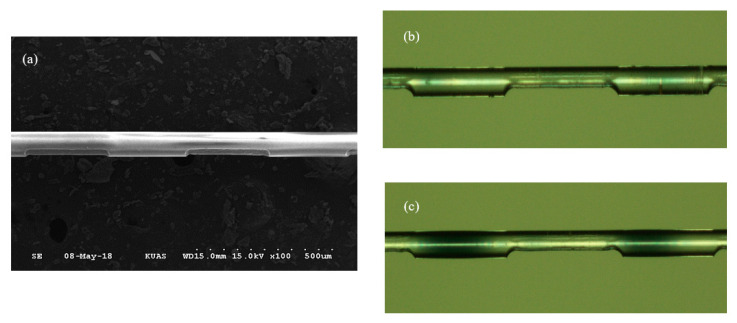
(**a**) SEM image of the optical fiber with the incident laser beam on the grating depth during wet etching; (**b**) microscan photo of the optical fiber with the incident laser beam on the grating depth during wet etching; (**c**) microscan photo of the optical fiber after polymer coating.

**Figure 3 polymers-12-02023-f003:**
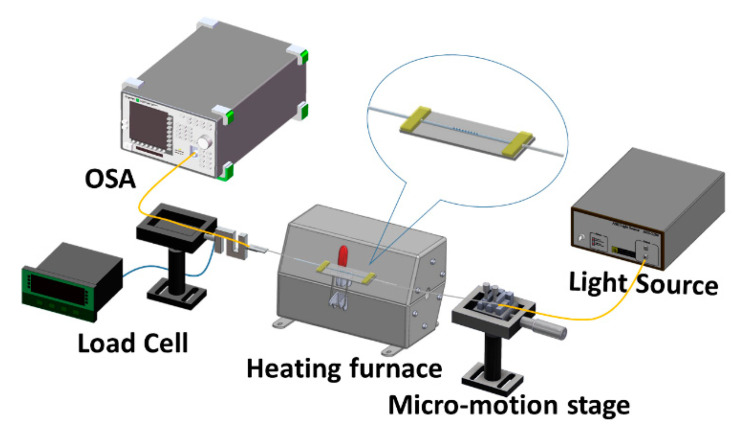
Schematic of the experimental setup for the long-period fiber grating (LPFG) sensor in the temperature detection experiment.

**Figure 4 polymers-12-02023-f004:**
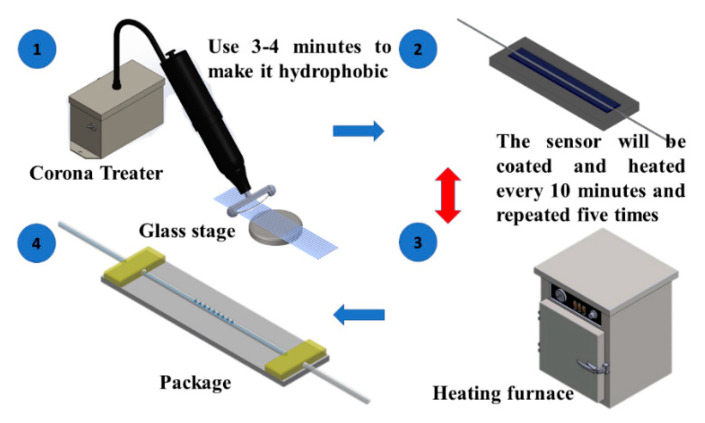
Schematic of the LPFG sensor detailing the process of the polymer coating.

**Figure 5 polymers-12-02023-f005:**
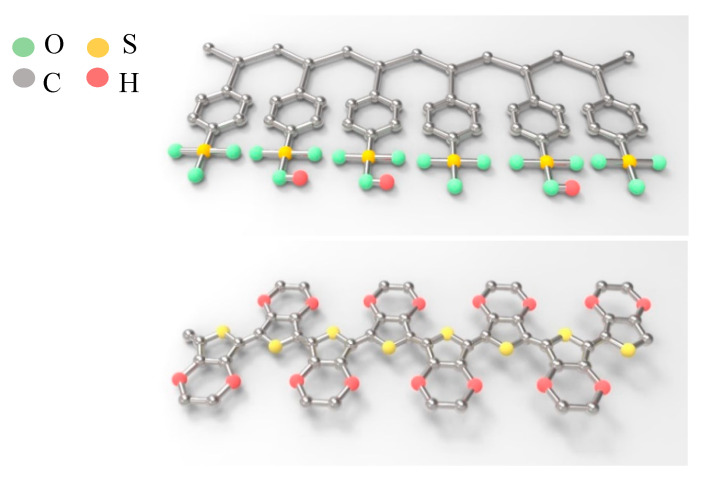
Schematic molecular structure of PEDOT:PSS.

**Figure 6 polymers-12-02023-f006:**
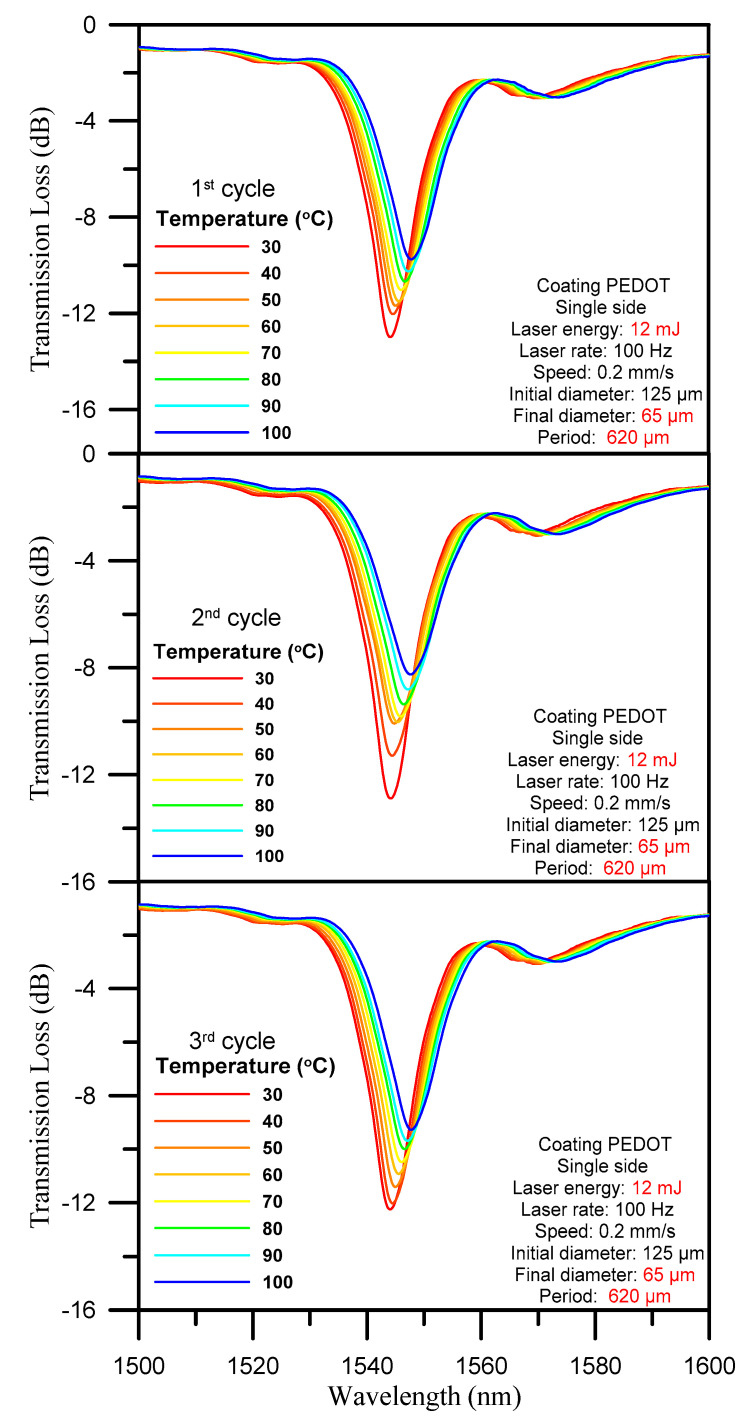
Three-cycle temperature test spectrogram with 10 °C intervals.

**Figure 7 polymers-12-02023-f007:**
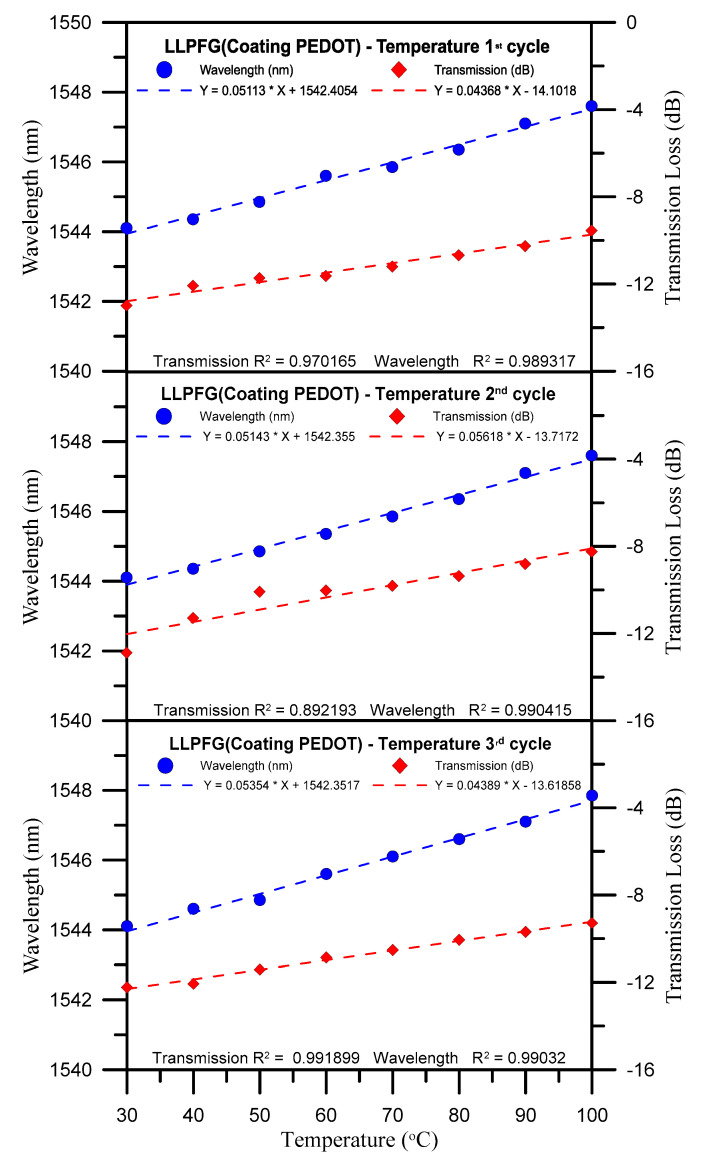
Chart of wavelength loss over three temperature test cycles.

**Figure 8 polymers-12-02023-f008:**
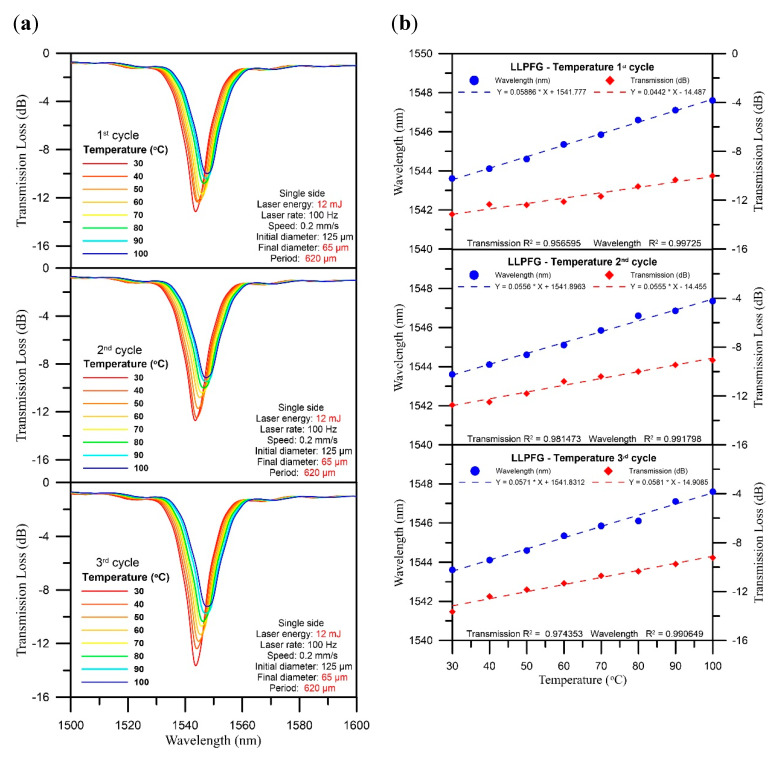
The LPFG sensor without PEDOT:PSS: (**a**) three-cycle temperature test spectrogram with 10 °C intervals; (**b**) chart of wavelength loss over three temperature test cycles.

**Table 1 polymers-12-02023-t001:** Changes in wavelength shift and transmission loss over three temperature cycles under LPFG (D = 65 μm, Λ = 620 μm).

Cycle	Wavelength (nm)	Transmission Loss (dB)
1st	3.54	−3.44
2nd	3.49	−4.63
3rd	3.49	−2.95

**Table 2 polymers-12-02023-t002:** Wavelength shift and transmission loss sensitivity over three temperature test cycles under LPFG (D = 65 μm, Λ = 620 μm).

Cycle	Wavelength Sensitivity (nm/°C)	Transmission Loss Sensitivity (dB/°C)
1st	0.0511	0.0437
2nd	0.0514	0.0562
3rd	0.0535	0.0439

**Table 3 polymers-12-02023-t003:** Linearity of wavelength shift and transmission loss over three temperature cycles under LPFG (D = 65 μm, Λ = 620 μm).

Cycle	Wavelength Linearity (R^2^)%	Transmission Loss Linearity (R^2^)%
1st	98.9317	97.0165
2nd	99.0415	89.2193
3rd	99.0320	99.1988

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
