# Peer review of "Long-Period Fiber Grating Sensor Based on a Conductive Polymer Functional Layer"

_polymers, 2020, doi:10.3390/polym12092023_

Round 1
Reviewer 1 Report
Manuscript No: polymers-906757
Title: Long-Period Fiber-Grating Sensor Based on a Conductive Polymer Functional Layer
Authors: Ching-Yu Hsu, Chia-Chin Chiang, Hsin-Yi Wen, Jian-Jie Weng, Jing-Lun Chen, Tao-Hsing Chen and Ya-Hui Chen
- Overview
- In this manuscript the authors present a laser-assisted wet-chemical etching process for controlling the grating depth of LPFG.
- The contents are not clearly expressed; the manuscript is not well organized and is written in poor English.
Look for typos.
- The authors have acknowledged recent related research.
- As long as my knowledge, the work presented is original but is not correct from a scientific point of view.
- Detailed analysis
- Keywords, please change to: laser-assisted-etching LPFG (LLPFG); long period fiber grating (LPFG); fiber sensor; …
- Abstract:
Please make it as synthetic as you can. It must be clear, objective and self-explanatory. State what have you done, how did you do it, the quantitative results you got. State briefly the originality of the work, What is new in the work?
Sentence beginning at Line 19 (to 22) is of no use here
- Introduction: to lengthy, it must be reduced. Be concise and objective.
- sentence Lines 28-31: useless, remove it
- sentence Lines 31 to 34: it does not make sense starting with generalities and then coupling theory.
- sentence Lines 35-36: useless, remove it
- sentence Lines 40-43: very confusing
- “the LPFG had photoperiod of 280 m within the wavelength range of 1.1–1.7 nm (0 °C to 100 °C).” It does not make sense.
- Delete the template Lines 106 to 113
- Theory
- sentence Lines 131-135: very confusing
- Process and experiment
This is Material and Methods
- Results and discussion
- Table 3: R-squared is always between 0 and 100%: this is unacceptable.
- For this work to make any sense you must measure the temperature sensitivity of a LPFG without PEDOT:PSS
- Overall assessment
In its present form the work reported presents no utility for studies and developments in the field.
Control experiments are missing.
In my opinion it must be rejected, unless the authors redesign the experiment.
- Review Criteria
- Scope of Journal
Rating: Moderately high
- Novelty and Impact
Rating: Low
- Technical Content
Rating: Low
- Presentation Quality
Rating: Low
Author Response
Dear Ms. Claire Zhang Assistant Editor,
Title: Long-Period Fiber-Grating Sensor Based on a Conductive Polymer
Functional Layer
Manuscript ID: polymers-906757
We greatly appreciated your comments and these of the reviewer. These comments have helped us to improve our manuscript considerably. Following this letter are the editor and reviewer comments with our responses, including how and where the text was modified. Changes made in the manuscript are marked using “the red font sections have been modified”. The revision has been developed in consultation with all coauthors, and each author has given approval to the final form of this revision. Please consider this paper for publication.
Sincerely,
Chia-Chin Chiang
Professor, National Kaohsiung University of Science and Technology
Email:ccchiang@nkust.edu.tw
TEL: +886-7-3814526 ext 15340
FAX: +886-7-3831373
Postal address: no. 415, Jiangong Rd., Sanmin Dist., Kaohsiung City 80778, Taiwan

Reviewer 2 Report
In this manuscript, Ching-Yu Hsu et. al, report the fabrication of long-period fiber grating (LPFG) using laser-assisted wet-chemical etching and coated PEDOT:PSS layer for flexible/bending application. They successfully fabricated the temperature sensor with LPFG optical fibers, and tested the sensor properties by measuring the wavelength shift and transmission loss as a function of temperature. The wavelength shift and transmission loss exhibited linearity as a function of temperature, which implies that the sensor could be used as a temperature sensor. Despite to their results, I have some question about their results. As they mentioned in the manuscript, the transmission loss of the sensor is a function of cos^2*(coupling coefficient x grating length L). When the grating length is changes due to the thermal expansion or contraction, the transmission loss should be changed. However, if the grating length is a function of temperature, i.e the length µ thermal expansion coefficient x temperature (it is simply assumed that the length changes µ thermal expansion coefficient x temperature), the transmission loss should be a function of cos^2(temperature), i.e it seems that the linearity of transmission loss (as a function of temperature) could not be expected in eq(3). Could you explain this?
In addition, in which temperature range do you expect the linearity of wavelength & transmission loss to break? Do you expect that the linearity works even at low temperature region? Finally, the transmission loss at 100 deg was ~10 at 1 cycle, -8 at 2nd cycles, ~9.x at 3rd cycles. The value from the 2nd cycle seems to be different to the 1 and 3 cycles. Could you explain why the 2nd cycle exhibited somewhat different behavior?
Author Response

(The authors gave the same response as above.)

Round 2
Reviewer 1 Report
Manuscript No: polymers-906757 R1
Title: Long-Period Fiber-Grating Sensor Based on a Conductive Polymer Functional Layer
Authors: Ching-Yu Hsu, Chia-Chin Chiang, Hsin-Yi Wen, Jian-Jie Weng, Jing-Lun Chen, Tao-Hsing Chen and Ya-Hui Chen
- Overview
- In this manuscript the authors present a laser-assisted wet-chemical etching process for controlling the grating depth of LPFG.
- The contents are not clearly expressed; the manuscript is well organized and is written in reasonable English.
- The authors have acknowledged recent related research.
- As long as my knowledge, the work presented is original and correct from a scientific point of view.
- Overview
The authors presented accurate responses to the Reviewers comments and queries and introduced the required changes in the manuscript.
The work reported presents reasonable utility for supplementary studies and developments in the field.
In my opinion it may be published.
- Review Criteria
- Scope of Journal
Rating: Medium
- Novelty and Impact
Rating: Medium
- Technical Content
Rating: Medium
- Presentation Quality
Rating: Medium